# Efficient Fine-tuning via Auxiliary Representation

## Abstract

The widespread adoption of large pretrained models has made fine-tuning an essential step for tailoring models to specific tasks. As these models continue to scale larger and as the demand for task-specific and personalized adaptation grows, parameter-efficient fine-tuning (PEFT) has emerged as a practical alternative to full fine-tuning. PEFT enables effective adaptation while updating only a small fraction of the total parameters. While various PEFT techniques have shown strong performance, many still suffer from increased inference latency and inefficiencies in multi-adapter scenarios. Motivated by these limitations, we propose a novel PEFT approach that leverages auxiliary representations to enable fast and flexible inference. In our method, **Lat**ent **T**ask **E**mbedding fine-tuning, a small task-specific latent embedding is concatenated to the original embedding. The corresponding weight matrices are extended, and only the additional parameters introduced by this expansion are trained. This design allows for efficient inference using a single matrix multiplication per weight, minimizing latency overhead, and supports task-specific masking to handle multiple adapters within a single model. We evaluate our method on large language models and latent diffusion models, demonstrating competitive accuracy with existing PEFT baselines while providing faster inference and enabling efficient intra-batch multi-task processing.

## 1 Introduction

The remarkable success of large pretrained models is largely attributed to scale and generality – large-scale training on diverse data results in highly capable models that can be fine-tuned for a wide range of downstream applications (Achiam et al., 2023; Grattafiori et al., 2024; Abdin et al., 2024; Yang et al., 2024; Team et al., 2025). However, as model sizes continue to grow, full fine-tuning (FFT), where all parameters are updated for each task, has become increasingly impractical due to its high computational and storage demands. To address this, parameter-efficient fine-tuning (PEFT) methods have emerged as a compelling alternative. PEFT techniques adapt pretrained models by introducing and updating only a small number of parameters while keeping the base model frozen. Various PEFT approaches have shown that strong downstream performance can be achieved with significantly fewer trainable parameters, enabling rapid adaptation and deployment (Houlsby et al., 2019; Lester et al., 2021; Hu et al., 2022).

While existing PEFT methods have proven effective, the rapidly expanding landscape of downstream applications – ranging from hyper-personalization (Chen et al., 2024) to privacy-preserving edge deployment (Xu et al., 2024) – poses critical challenges for inference efficiency. In these resource-constrained environments, where dedicated cloud-tier GPUs are often absent, the demand extends beyond simple adaptation to the simultaneous serving of multiple task-specific adapters. This requirement exposes a fundamental limitation in current approaches: a stark trade-off between latency and memory. Merging adapters (e.g., via LoRA) achieves fast single-task inference but forces memory usage to scale linearly with the number of tasks, rendering it impractical for multi-tenant scenarios. Conversely, maintaining unmerged adapters conserves memory but introduces sequential computational overhead that significantly degrades latency. Consequently, modern deployment faces an unresolved trilemma among accuracy, latency, and multi-task flexibility that existing methods struggle to resolve.

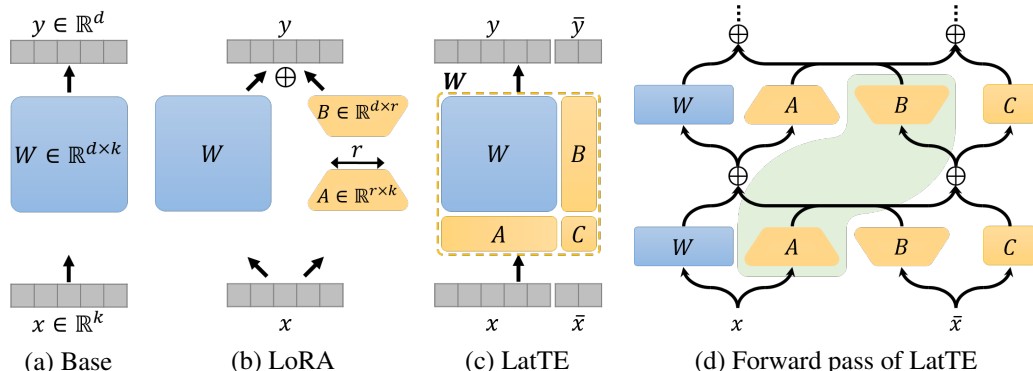

Figure 1: (a-c) The comparison of the forward pass through a single weight for the base, LoRA, and LatTE model. While the $A$ and $B$ matrices in LoRA are expressed conventionally as trapezoids to emphasize their low ranks, they are identical in the matrix dimensions with the $A$ and $B$ in LatTE. (d) The forward pass a MLP model with LatTE. We have omitted the non-linearity between the layers for simplicity. The shaded region can be conceptually regarded as a single LoRA unit.

PEFT techniques can be broadly categorized into three types. The first modifies the model architecture, such as in serial or parallel adapters (Houlsby et al., 2019; Pfeiffer et al., 2021; He et al., 2022), by introducing adapter modules into each transformer block which alters the flow of intermediate representations. The second category centers on finding efficient ways to tune the weight matrices that resemble FFT. This category includes LoRA (Hu et al., 2022) and its variants. The third category focuses on representations, and either introduce additional learnable embeddings such as prompt-based methods (Lester et al., 2021; Li & Liang, 2021; Liu et al., 2022b), or modify the hidden states to steer the model's behavior (Liu et al., 2022a; Wu et al., 2024b). While each strategy has its merits, existing PEFT approaches do not focus on inference efficiency.

In this work, we propose a new direction: leveraging *auxiliary latent representations* as a compact and efficient carrier of task-specific information. This design choice offers several distinct advantages: (1) it preserves fast inference by reducing the adaptation to the same single matrix multiplication as in the base model (but with increased dimension), (2) its simple architecture introduces few hyperparameters and has low memory requirement, and (3) it enables flexible composition and switching of multiple tasks. Motivated by these properties, we introduce **Lat**ent **T**ask **E**mbedding (**LatTE**) fine-tuning, a novel PEFT method that injects task-specific latent embeddings directly into the model's input layer. LatTE enables fast, simple, and composable fine-tuning with minimal latency overhead and strong empirical performance across diverse tasks and model families.

LatTE concatenates the learnable auxiliary embedding to the original latent embedding, which serves as a compact, task-specific representation. The projection weights are expanded accordingly, and only the newly-introduced parameters are updated during fine-tuning. The gist of the method, acting on a single weight, is compared with the base model and LoRA in Figure 1 (a-c). This results in a lightweight and latency-free tuning mechanism, where inference remains as efficient as the base model. Our approach enables multiple adapters to coexist within the enlarged embedding space with task-specific mask controlling the adapter to be used – an important capability for efficient multi-task and multi-domain deployment. We evaluate our method across various LLMs and a diffusion model on a variety of tasks. Both experimental results and theoretical analysis demonstrate that LatTE consistently matches the performance of leading PEFT baselines while achieving faster inference speed. By bridging the gap between efficient fine-tuning and real-world deployment constraints, our method paves the way for scalable and high-performance model adaptation in practical applications.

We summarize our main contributions as follows:

- We propose LatTE, a novel PEFT method that utilizes auxiliary latent embeddings, enabling low-latency adaptation with a minimal number of trainable parameters.
- We conduct extensive experiments across LLMs and diffusion models, demonstrating strong performance compared to robust PEFT baselines.

- We demonstrate the unique flexibility of LatTE for efficient intra-batch multi-task inference via task-specific masking, maintaining near-constant latency when serving heterogeneous tasks simultaneously.

## 2 RELATED WORKS

**Parameter-Efficient Fine-Tuning (PEFT).** PEFT is a series of methods that update only a small fraction of the model's parameters during fine-tuning, while keeping the rest frozen. This enables adaptation to new tasks with minimal computational overhead while preserving the capabilities of the base model. We broadly categorize existing PEFT methods into the following three classes.

- **Module-based methods** introduce additional trainable modules (adapters) to the model architecture. The adapters mostly consist of a down- and up-projection with a nonlinearity in between. There are serial adapter (Houlsby et al., 2019; Pfeiffer et al., 2021) and parallel adapter (He et al., 2022) based on how the adapter is attached to the base module. CIAT (Zhu et al., 2021) and CoDA (Lei et al., 2023) are also variants of the parallel adapter.
- **Weight-based methods** aim to efficiently update the weights of the model, and can be considered as a direct approximation of FFT. The update of the weights can be either additive or multiplicative. LoRA (Hu et al., 2022) and its variants (Zhang et al., 2023b; Liu et al., 2024a; Kopiczko et al., 2024; Li et al., 2024), the most widely used PEFT, are additive weight-based methods. Multiplicative-update method includes OFT (Qiu et al., 2023), BOFT (Liu et al., 2024b), and HRA (Yuan et al., 2024).
- **Representation-based methods** use representations as a tool for fine-tuning. Soft-prompt methods are in this category, where learnable embeddings are prefixed to the prompt. Prefix-tuning (Li & Liang, 2021), prompt-tuning (Lester et al., 2021), and p-tuning (Liu et al., 2022b) are examples of such methods. Another group in this category modifies or edits the intermediate representations to fit the model to downstream tasks. This includes methods such as $(IA)^3$ (Liu et al., 2022a), SSF (Lian et al., 2022), and ReFT (Wu et al., 2024b).
- **Sparsity-Based methods** leverages sparsity and selectively update subsets of model parameters identified as most critical for the target task, often by masking (Guo et al., 2021; Sung et al., 2021; Ansell et al., 2022). More recent work on sparsity methods, SpIEL (Ansell et al., 2024) and SMT (He et al., 2025), scales these ideas to LLMs up to 13B parameters.

Notably, despite the extensive research over the past years, no PEFT approach explores the use of representations auxiliary to the embeddings as a mechanism for adaptation, to the best of our knowledge.

**Inference-Efficient PEFT.** Recent works have recognized the need for reducing inference-time overhead in PEFT. SPLoRA (Hedegaard et al., 2024) and CA-LoRA (Zhao et al., 2024) combine LoRA with pruning or quantization for faster inference. Our method differs from these approaches as we preserve the precision and expressivity of the base model. Liao et al. (2023) introduces zero latency PEFT methods, PaFi and HiWi, which are a task-agnostic sparse fine-tuning and a multiplicative version of LoRA, respectively. These methods do not introduce additional latency as they (partially) update the original model as in FFT. Inference of LatTE, on the other hand, is as fast as the base model while keeping the original weights intact, which is beneficial for fast task-switching.

We emphasize a critical distinction often overlooked in the literature: many weight-based methods such as LoRA and OFT claim inference speed identical to the base model. However, this only holds when weight updates are pre-merged before inference – i.e., low-rank weights are pre-added for LoRA or orthogonal matrices are pre-multiplied for OFT. Pre-merging destroys multi-task capability: serving $N$ tasks requires $N$ separate merged models or constant loading/unloading overhead. In contrast, LatTE maintains near-base model inference speed while supporting multiple tasks through a single expanded model with task-specific masking, making it uniquely suited for multi-adapter deployment scenarios.

**Multi-Adapter and Multi-Task Adaptation.** Scenarios involving multiple downstream tasks or domains often demand flexible and composable fine-tuning strategies. MAD-X (Pfeiffer et al., 2020)

and AdapterFusion (Pfeiffer et al., 2021) allow for combining multiple adapters, yet require additional merging logic and may incur runtime costs. Mixture of Expert (MoE) (Jacobs et al., 1991) style combination of multiple task LoRAs (Wu et al., 2024a; Xu et al., 2025; Liao et al., 2025) may be effective, but adds complexity during inference. Our approach supports multiple tasks through task-specific masking applied to the embedding, introducing minimal latency compared to single-task inference. Moreover, this flexibility is an intrinsic feature of vanilla LatTE, and has much room for improvements in future variants specialized for such purposes.

# 3 OUR METHOD

We propose LatTE, a fine-tuning method in which a task-specific auxiliary embedding is concatenated to the original token embeddings. This increases the embedding dimensionality, which correspondingly expands the associated weight matrices. The fine-tuned knowledge is thus stored in the additional weight parameters introduced by this expansion, and the interaction with the original forward pass is mediated by the auxiliary embedding. Architecturally, LatTE resembles LoRA and other weight-based PEFT methods, as it modifies individual weights without altering the model's overall structure (see Figure 1). However, a unique feature of LatTE is that the forward pass through one weight is calculated by a single matrix multiplication while keeping the original weight frozen. This enables inference-time performance comparable to that of the original model, making LatTE both efficient and scalable.

The comparison between the base model, LoRA, and LatTE is illustrated in Figure 1 (a-c), which shows the forward pass through a single weight matrix $W \in \mathbb{R}^{d \times k}$. Given an input embedding $x \in \mathbb{R}^k$, the base model computes the output $y \in \mathbb{R}^d$ as $y = Wx$. In LoRA, task-specific information is introduced via a low-rank adapter, yielding: $y = Wx + BAx$, where $A \in \mathbb{R}^{r \times k}$ and $B \in \mathbb{R}^{d \times r}$, and $r$ is the low-rank dimension. In LatTE, the input embedding is concatenated with an auxiliary embedding $\bar{x} \in \mathbb{R}^r$ to form $[x; \bar{x}] \in \mathbb{R}^{k+r}$. The forward pass is computed in the expanded embedding space:

$$\begin{bmatrix} y \\ \bar{y} \end{bmatrix} = \begin{bmatrix} W & B \\ A & C \end{bmatrix} \begin{bmatrix} x \\ \bar{x} \end{bmatrix}, \tag{1}$$

where $A$, $B$, and $C \in \mathbb{R}^{r \times r}$ are trainable matrices. For convenience, we denote the expanded vectors and matrix as $\mathbf{x} = \begin{bmatrix} x \\ \bar{x} \end{bmatrix}$, $\mathbf{y} = \begin{bmatrix} y \\ \bar{y} \end{bmatrix}$, and $\mathbf{W} = \begin{bmatrix} W & B \\ A & C \end{bmatrix}$, so that Eq. (1) simplifies to the familiar linear form $\mathbf{y} = \mathbf{W}\mathbf{x}$. In Figure 1 (c), the block matrix $\mathbf{W}$ is highlighted with a yellow dashed outline. We deliberately set the auxiliary embedding dimension $r$ to match the low-rank dimension used in LoRA, allowing for a fair parameter comparison. LoRA introduces $(d + k)r$ additional parameters per weight matrix, while LatTE adds $(d + k + r)r$, which remains comparable under the common assumption $r \ll \min(d, k)$. Proper initialization is crucial to PEFT model's performance and we initialize $A$ as Kaiming uniform and $B$, $C$ to zero. This follows LoRA's default strategy in the Huggingface PEFT library (Mangrulkar et al., 2022), and ensures that LatTE's forward pass exactly replicates the base model at the start of the training.

To compute in the expanded space, we define expansion and compression functions $f_\text{in}$ and $f_\text{out}$, which are applied once per forward pass:

$$f_\text{in}(x) = \mathbf{x}, \quad f_\text{out}(\mathbf{y}) = y.$$

In practice, we initialize the auxiliary embedding a learnable constant vector $c$: $f_\text{in}(x) = [x; c]$, and extract the final output as a linear combination: $f_\text{out}([y; \bar{y}]) = y + B\bar{y}$. Note that the matrix $B$ used in this post-processing step is a separate trainable parameter and not reused from the block matrix $\mathbf{W}$, although it shares the same shape $\mathbb{R}^{d \times r}$. Alternative design choices for $f_\text{in}$ and $f_\text{out}$ are discussed in Section 5. We emphasize that $f_\text{in}$ and $f_\text{out}$ are applied only once per forward pass through the entire network, not once per layer. The auxiliary embedding is expanded once at the input ($f_\text{in}$), propagates through all layers in the expanded space, and is compressed once at the output ($f_\text{out}$), ensuring minimal overhead.

## 3.1 APPLICATION TO THE MULTI-LAYER PERCEPTRON

With the LatTE building block in place, we now apply it to a multilayer perceptron (MLP) network. Consider an $L$-layer MLP with weight matrices $W_i$ for $1 \le i \le L$ and an activation function $\sigma$.

Given an input embedding $x$, the MLP computes the output as:

$$y = W_L \cdot \sigma(W_{L-1} \cdot \sigma(\cdots \sigma(W_1 x))).$$

To incorporate LatTE, we replace each $W_i$ with its expanded counterpart $\mathbf{W}_i$ and insert the expansion and compression functions $f_{\text{in}}$ and $f_{\text{out}}$ at the input and output, respectively. The input embedding is first expanded from a $d$-dimensional vector to a $(d + r)$-dimensional one by $f_{\text{in}}$, then propagated through the network using the sequence of $\mathbf{W}_i$ matrices. The final embedding is compressed back to a $d$-dimensional output by $f_{\text{out}}$. The resulting output becomes:

$$y = f_{\text{out}}\left(\mathbf{W}_L \cdot \sigma(\mathbf{W}_{L-1} \cdot \sigma(\cdots \sigma(\mathbf{W}_1 f_{\text{in}}(x))))\right). \tag{2}$$

The feed-forward network (FFN) block in a Transformer architecture (Vaswani et al., 2017) is typically a two-layer MLP of the form:

$$\text{FFN}_\sigma(x, W_1, W_2) = W_2 \cdot \sigma(W_1 x).$$

Modern LLMs, including those used in our experiments (Grattafiori et al., 2024; Yang et al., 2024), often use the SwiGLU activation function (Shazeer, 2020), a variant of the Gated Linear Unit (GLU) (Dauphin et al., 2017) based on Swish (Ramachandran et al., 2017). A single FFN using SwiGLU can be expressed as:

$$\text{FFN}_{\text{SwiGLU}}(x, W, V, W_2) = W_2 \cdot (\text{Swish}_1(Wx) \otimes Vx),$$

where $\text{Swish}_\beta(x) = x \cdot \text{sigmoid}(\beta x)$ and $\otimes$ denotes the element-wise (Hadamard) product. Applying LatTE to this FFN block involves replacing each weight matrix with its expanded form. The resulting forward pass becomes:

$$\text{FFN}_{\text{SwiGLU}}(\mathbf{x}, \mathbf{W}, \mathbf{V}, \mathbf{W}_2) = \mathbf{W}_2 \cdot (\text{Swish}_1(\mathbf{W}\mathbf{x}) \otimes \mathbf{V}\mathbf{x}). \tag{3}$$

### 3.2 APPLICATION TO THE ATTENTION MODULE

Unlike MLP layers – where applying LatTE is straightforward and leaves little room for design choices beyond $f_{\text{in}}$ and $f_{\text{out}}$ – the attention module presents several implementation options. The core strategy remains the same: expand the embedding dimension and replace the weight matrices with their extended counterparts. However, the key challenge lies in how to allocate the auxiliary embedding dimensions across the attention heads.

Assume a standard multi-head attention (MHA) mechanism with $H$ heads, where the per-head dimension is $d_H = d/H$. The usual output of an MHA layer can be written as:

$$\sum_{h=1}^{H} (W_h^O)^\top W_h^V x \cdot \text{softmax}\left((W_h^K x)^\top W_h^Q x\right),$$

where $W_h$ denotes the $h$-th row-wise split submatrix of a full weight matrix $W$, i.e., $W_h = W[(h-1)d_H : hd_H, :]$. When the embedding dimension increases from $d$ to $d + r$, we must adjust the MHA configuration so that $d_H = (d + r)/H$ holds. Two simple approaches can achieve this:

1. **More heads**: Increase the number of attention heads $H$ such that the added auxiliary dimension $r$ fits into additional heads. Specifically, $\lceil r/d_H \rceil$ additional heads are introduced.

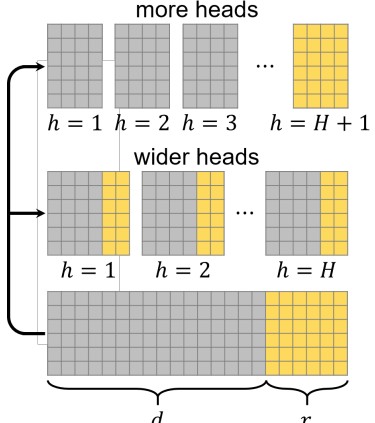

more heads

$h=1$  $h=2$  $h=3$  $\cdots$  $h=H+1$

wider heads

$h=1$  $h=2$  $\cdots$  $h=H$

$\underbrace{\qquad}_{d}$  $\underbrace{\quad}_{r}$

Figure 2: The two strategies on allocating the additional dimension of $Wx$ (yellow) to attention heads. Here, the vertical dimension represents the context length.

2. **Wider heads**: Increase the per-head dimension $d_H$ while keeping the number of heads fixed. This distributes the auxiliary dimension across all heads by enlarging $d_H$ by $2\lceil r/(2H) \rceil$. The factor of 2 ensures an even extension, which is necessary to preserve compatibility with rotary positional embeddings (Su et al., 2024), where embedding vectors are decomposed into pairs for $2d$-rotations.

If $r$ is not divisible by $d_H$ (the more-heads case) or by $H$ (the wider-heads case), the unmatched dimensions are zero-padded. Figure 2 provides a schematic of these two integration strategies.

| Model | Fintuning | Params (%) | Accuracy | | | | | | | | |
| --- | --- | --- | --- | --- | --- | --- | --- | --- | --- | --- | --- |
| | | | OBQA | BoolQ | PIQA | SIQA | Hella. | Wino. | ARC-e | ARC-c | ave. |
| Qwen2.5-3B | Prompt | 0.0021 | 77.04 | 80.46 | 80.60 | 73.54 | 69.95 | 62.63 | 93.82 | 82.98 | 77.63 |
| | Serial | 0.7546 | 86.70 | 86.86 | 85.81 | 79.20 | 91.99 | 79.99 | 93.39 | 83.53 | 85.93 |
| | Parallel | 0.7546 | 85.90 | 87.08 | 85.77 | 79.77 | 92.26 | 82.75 | 94.03 | 83.32 | 86.36 |
| | OFT | 0.7025 | 87.50 | 87.58 | 85.69 | 78.97 | 92.36 | 82.44 | 93.98 | 83.49 | 86.50 |
| | BOFT | 0.7025 | **87.90** | 87.74 | **86.29** | **79.86** | 92.01 | 82.06 | 93.81 | 83.21 | 86.61 |
| | LoRA | 0.7546 | 87.15 | 87.50 | 86.28 | 79.38 | **92.46** | 83.11 | 93.61 | 83.29 | 86.60 |
| | **LatTE-m** | 0.7573 | 86.04 | 87.60 | 85.58 | 79.02 | 91.98 | **84.35** | 93.60 | 83.12 | 86.41 |
| | **LatTE-w** | 0.7573 | 86.80 | **87.97** | 85.73 | 78.67 | 92.20 | 83.94 | **94.14** | **83.55** | **86.63** |
| Qwen2.5-7B | Prompt | 0.0015 | 86.50 | 85.85 | 86.04 | 76.67 | 81.59 | 68.01 | 96.20 | 89.57 | 83.81 |
| | Serial | 0.5273 | **92.84** | 89.45 | 89.40 | 81.88 | 94.03 | 85.83 | 96.19 | 89.18 | 89.85 |
| | Parallel | 0.5273 | 92.04 | 89.30 | 89.15 | 81.26 | 94.01 | 87.83 | 96.50 | 89.76 | 89.98 |
| | OFT | 0.4943 | 91.84 | 89.04 | 90.08 | 80.99 | 93.84 | 85.65 | 96.68 | 89.80 | 89.74 |
| | BOFT | 0.4943 | 92.35 | 89.46 | 89.67 | 79.94 | 94.14 | 86.15 | 96.39 | 89.69 | 89.72 |
| | LoRA | 0.5303 | 92.55 | 89.59 | 90.22 | **81.94** | 93.84 | 86.76 | 96.57 | 90.29 | 90.22 |
| | **LatTE-m** | 0.5317 | 91.84 | **89.81** | 90.08 | 81.28 | 94.52 | 87.49 | 96.74 | **90.76** | 90.32 |
| | **LatTE-w** | 0.5317 | 92.60 | 89.49 | **90.59** | 80.73 | **94.67** | **87.86** | **97.00** | 90.25 | **90.40** |
| Llama-3.2-3B | Prompt | 0.0031 | 71.64 | 79.30 | 77.20 | 67.41 | 62.12 | 60.10 | 85.47 | 72.54 | 71.97 |
| | Serial | 0.7550 | 82.10 | 86.73 | 84.22 | 77.19 | 89.61 | 80.84 | 88.90 | 77.52 | 83.39 |
| | Parallel | 0.7550 | 82.10 | 87.50 | 83.42 | 77.79 | 90.66 | 82.57 | 89.27 | 78.65 | 83.99 |
| | OFT | 0.7666 | 84.84 | 88.44 | 85.01 | 79.17 | 90.91 | 82.89 | **91.25** | **79.82** | 85.29 |
| | BOFT | 0.7666 | 86.30 | **88.76** | 85.55 | 78.51 | 91.32 | 81.77 | 90.88 | 78.82 | 85.24 |
| | LoRA | 0.7568 | 84.44 | 88.56 | **85.61** | 77.94 | 91.41 | 83.78 | 89.82 | 78.07 | 84.95 |
| | **LatTE-m** | 0.7599 | 85.70 | 88.35 | 85.36 | 78.39 | **91.42** | **84.89** | 89.98 | 79.24 | 85.42 |
| | **LatTE-w** | 0.7599 | **86.40** | 88.74 | 85.23 | **79.21** | 91.15 | 84.47 | 90.49 | 79.76 | **85.68** |
| Llama-3.1-8B | Prompt | 0.0016 | 82.35 | 86.12 | 84.48 | 77.38 | 83.29 | 72.26 | 92.56 | 80.87 | 82.41 |
| | Serial | 0.4570 | 89.70 | 88.73 | 87.81 | 80.45 | 93.61 | 84.35 | 94.00 | 83.92 | 87.82 |
| | Parallel | 0.4570 | 89.75 | 88.87 | 88.15 | 81.35 | 94.10 | 86.03 | 93.80 | **84.91** | 88.37 |
| | OFT | 0.4492 | 89.00 | 89.30 | 88.45 | **81.54** | 93.79 | 84.02 | 94.43 | 84.68 | 88.15 |
| | BOFT | 0.4492 | 88.90 | **89.72** | 88.75 | 80.81 | 93.58 | 84.85 | 94.32 | 83.94 | 88.11 |
| | LoRA | 0.4570 | 88.90 | 89.09 | 88.85 | 80.92 | 94.31 | 87.23 | 93.81 | 84.85 | 88.50 |
| | **LatTE-m** | 0.4585 | 90.15 | 89.63 | 88.80 | 80.80 | **94.39** | **88.38** | 94.14 | 84.45 | **88.84** |
| | **LatTE-w** | 0.4585 | **90.60** | 89.33 | **88.87** | 80.28 | 94.25 | 86.68 | **94.73** | 84.47 | 88.65 |

Table 1: Accuracy results on the commonsense QA benchmark, which includes eight diverse reasoning tasks. Adapters are applied to all layers.

With the application of LatTE to both FFN and attention layers explained, we are now ready to implement it in Transformer-based LLMs. In diffusion models, PEFT is typically applied to the text encoder and the cross-attention modules within the U-Net architecture (Ruiz et al., 2023; Zhang et al., 2023a; Mou et al., 2024). This means that the same recipe used for LLMs can be applied to diffusion models as well.

## 3.3 COMPARISON WITH LORA

We provide theoretical context for LatTE by drawing comparisons with LoRA. The intuition behind LoRA stems from the observation that the intrinsic dimension of many NLP tasks is significantly lower than the dimension of large pretrained models (Aghajanyan et al., 2021). Building on this, LoRA hypothesizes that the required weight updates for task adaptation also lie in a low-rank subspace (Hu et al., 2022). However, the argument in Aghajanyan et al. (2021) supports a low intrinsic dimension as a *necessary*, but not *sufficient*, condition for low-rank adaptation to be effective.

Interestingly, LatTE can also be interpreted as a composition of low-rank updates. As illustrated in Figure 1 (d), when examining the green-shaded information flow over two layers (omitting activation for simplicity), the input embedding $x$ is transformed by a low-rank operation $BA$ and reintegrated with the base embedding. Thus, while LoRA applies a single low-rank update per layer, LatTE effectively performs two low-rank transformations across two layers. This structural similarity suggests that the same theoretical motivation underlying LoRA – namely, that task-specific transformations can be captured in low-rank subspaces – also supports the design of LatTE.

Beyond the qualitative arguments presented above, we provide two formal results establishing cases where the expressive power of LatTE is equivalent to that of LoRA.

**Theorem 1** For linear models, the minimum low-rank dimension required for adapter models to exactly recover the FFT target is identical for both LoRA and LatTE.

| Model | Fintuning | Params (%) | Accuracy | | | | | |
|---|---|---|---|---|---|---|---|---|
| | | | AOuA | GPQA | MATH | GSM8K | SVAMP | ave. |
| Qwen2.5-1.5B | Prompt | 0.0016 | 37.80 | 24.24 | 37.80 | 54.80 | 72.33 | 45.39 |
| | Serial | 0.2982 | 42.13 | 25.25 | 44.20 | 65.00 | 71.67 | 49.29 |
| | Parallel | 0.2982 | 50.79 | 29.29 | 45.00 | 63.00 | 70.67 | 51.75 |
| | OFT | 0.2847 | 40.09 | 30.81 | 39.40 | 52.40 | 64.00 | 46.14 |
| | BOFT | 0.2847 | 41.34 | 28.79 | 39.60 | 54.00 | 68.00 | 46.35 |
| | LoRA | 0.2991 | 49.61 | 31.82 | 42.40 | 61.80 | 74.00 | 51.92 |
| | **LatTE-m** | 0.3003 | 50.79 | 31.31 | 40.80 | 62.60 | 73.67 | **51.83** |
| | **LatTE-w** | 0.3003 | 49.61 | 30.30 | 43.20 | 63.80 | 73.00 | **51.98** |
| Qwen2.5-3B | Prompt | 0.0011 | 46.85 | 25.76 | 49.20 | 67.00 | 70.67 | 51.89 |
| | Serial | 0.1877 | 49.61 | 26.26 | 52.80 | 76.40 | 78.67 | 56.75 |
| | Parallel | 0.1877 | 48.43 | 28.79 | 49.60 | 73.00 | 83.33 | 56.63 |
| | OFT | 0.1821 | 43.31 | 22.73 | 49.00 | 65.40 | 72.67 | 50.62 |
| | BOFT | 0.1821 | 45.67 | 28.79 | 48.20 | 65.60 | 71.00 | 51.85 |
| | LoRA | 0.1886 | 45.28 | 28.28 | 51.40 | 73.60 | 81.33 | 55.98 |
| | **LatTE-m** | 0.1894 | 49.61 | 30.30 | 53.20 | 71.00 | 80.33 | **56.89** |
| | **LatTE-w** | 0.1894 | 48.03 | 26.26 | 52.40 | 73.80 | 78.33 | **55.77** |
| Llama-3.2-1B | Prompt | 0.0027 | 21.46 | 25.25 | - | 36.10 | 53.50 | 34.08 |
| | Serial | 0.3991 | 18.11 | 25.00 | - | 37.10 | 55.33 | 33.89 |
| | Parallel | 0.3991 | 31.50 | 36.77 | - | 36.70 | 50.67 | 36.41 |
| | OFT | 0.3979 | 25.20 | 24.49 | - | 34.10 | 52.67 | 34.11 |
| | BOFT | 0.3979 | 25.98 | 20.96 | - | 34.40 | 52.17 | 33.38 |
| | LoRA | 0.3991 | 22.24 | 24.24 | - | 37.70 | 56.00 | 35.05 |
| | **LatTE-m** | 0.4009 | 24.80 | 22.22 | - | 38.10 | 57.17 | **35.57** |
| | **LatTE-w** | 0.4009 | 24.41 | 22.22 | - | 38.50 | 56.00 | 35.28 |
| Llama-3.2-3B | Prompt | 0.0015 | 54.92 | 23.74 | 40.60 | 71.20 | 78.83 | 53.86 |
| | Serial | 0.1874 | 51.57 | 30.56 | 37.10 | 69.10 | 78.83 | 53.43 |
| | Parallel | 0.1874 | 51.38 | 27.78 | 39.10 | 67.70 | 81.33 | 53.46 |
| | OFT | 0.1977 | 48.62 | 28.54 | 39.60 | 69.90 | 81.33 | 53.60 |
| | BOFT | 0.1977 | 50.59 | 28.54 | 37.20 | 68.00 | 78.67 | 52.60 |
| | LoRA | 0.1892 | 50.59 | 27.78 | 39.30 | 68.10 | 78.17 | 52.79 |
| | **LatTE-m** | 0.1902 | 45.67 | 26.26 | 39.10 | 67.70 | 79.67 | 51.68 |
| | **LatTE-w** | 0.1902 | 45.47 | 30.56 | 39.70 | 69.30 | 81.50 | 53.31 |

Table 2: Accuracy on multiple-choice and arithmetic reasoning benchmarks using various PEFT methods. MATH results for Llama-1B is omitted as it replied in forms which cannot be parsed in all methods.

**Theorem 2.** Any attention matrix produced by LoRA can be represented as an attention matrix from LatTE, given that $\bar{x}$ is a linear transform of $x$.

While rigorously establishing equivalence between highly non-linear models remains challenging, these results provide theoretical evidence supporting LatTE's expressive capacity. Complete proofs and a comprehensive theoretical analysis of LatTE's expressive power (Zeng & Lee, 2024) are provided in Appendix A.

## 4 EXPERIMENTS

We evaluate our LatTE method on both natural language processing (NLP) and text-to-image (T2I) generation tasks. For NLP, we conduct experiments using the Llama 3 (Grattafiori et al., 2024) and Qwen2.5 (Yang et al., 2024) language models, covering a range of model sizes from 1B to 8B parameters. For T2I generation, we fine-tune Stable Diffusion v1.5 (Rombach et al., 2022) as the base model. All fine-tuning is performed on 4 NVIDIA H100 GPUs, and inference is conducted using a single H100.

We compare both variants of LatTE – more heads (LatTE-m) and wider heads (LatTE-w) – against several strong PEFT baselines. The baseline methods include prompt tuning (Lester et al., 2021), serial (Houlsby et al., 2019) and parallel (He et al., 2022) adapters, OFT (Qiu et al., 2023), BOFT (Liu et al., 2024b), and LoRA (Hu et al., 2022). We follow the training setups from the respective references and use official implementations from the Huggingface PEFT library (Mangrulkar et al., 2022) for prompt tuning, LoRA, OFT, and BOFT.

Unless otherwise stated, we set the rank $r = 16$ for both LatTE and LoRA. For a fair comparison, the adapter hidden size and the block sizes for OFT and BOFT are chosen such that the number of trainable parameters closely match that of LatTE and LoRA; we use 2 blocks for BOFT by default.

| Design | Accuracy (Qwen2.5-7B) | | | | | | | | |
|---|---|---|---|---|---|---|---|---|---|
| | OBQA | BoolQ | PIQA | SIQA | Hella. | Wino. | ARC-e | ARC-c | ave. |
| Baseline (LatTE-w) | 92.60 | 89.49 | **90.59** | 80.73 | **94.67** | **87.86** | **97.00** | 90.25 | **90.40** |
| LatTE-m | 91.84 | **89.81** | 90.08 | 81.28 | 94.52 | 87.49 | 96.74 | **90.76** | 90.32 |
| $f_{\text{in}}(x) = [x; \mathbf{0}_r]$ | 90.53 | 96.56 | 89.50 | 92.00 | 89.61 | 80.73 | 94.40 | 86.50 | 89.98 |
| $f_{\text{out}}([y, \bar{y}]) = y$ | 90.19 | 96.77 | 89.86 | 91.56 | 89.89 | 81.12 | 93.83 | 85.78 | 89.88 |
| $f_{\text{in}}(x) = [x; \mathbf{0}_r]; f_{\text{out}}([y, \bar{y}]) = y$ | 90.44 | 96.72 | 89.77 | 92.40 | 89.69 | 80.30 | 94.27 | 86.44 | 90.00 |

Table 3: Accuracy results on the commonsense QA benchmark compared across different design choices, including $f_{\text{in}}$, $f_{\text{out}}$, and the more/wider heads.

All models are trained for up to 20 epochs for commonsense QA and 4 epochs for reasoning, and we report results using the best checkpoint selected via measuring with 4 different seeds.

## 4.1 COMMONSENSE QA

We begin with commonsense QA, a multiple-choice question answering task. Models are fine-tuned on the Commonsense 170K dataset (Hu et al., 2023), which comprises eight distinct QA benchmarks: OBQA (Mihaylov et al., 2018), BoolQ (Clark et al., 2019), PIQA (Bisk et al., 2020), SIQA (Sap et al., 2019), HellaSwag (Zellers et al., 2019), WinoGrande (Sakaguchi et al., 2021), ARC-easy, and ARC-challenge (Clark et al., 2018). These datasets focus on direct-answer selection without requiring chain-of-thought (CoT) reasoning (Wei et al., 2022).

Table 1 presents the performance of Llama-3.2-3B, Llama-3.1-8B and Qwen2.5-3B/7B models fine-tuned with LatTE and baseline methods. Across all model sizes and configurations, LatTE consistently matches or exceeds the performance of established PEFT baselines. Notably, LatTE-w achieves the highest average performance on three of the four models and ranks second on the remaining model. This performance gain exceeds our theoretical expectation, which suggested LatTE would perform comparably to LoRA. While further investigation is needed, we hypothesize that the $C$ matrix contributes significantly to this improvement. We also observe that LatTE demonstrates superior training efficiency relative to baseline methods, achieving better performance within a single epoch.

## 4.2 MULTIPLE-CHOICE AND ARITHMETIC REASONING

We next evaluate our method on reasoning tasks. These tasks additionally require chain-of-thought (CoT) reasoning before arriving at a final answer. We construct the training set by filtering examples from the Llama-Nemotron-Post-Training-Dataset (Bercovich et al., 2025), selecting those whose answers contain numeric characters and do not involve blank spaces. This filtering yields approximately 0.3M examples per epoch for training.

We evaluate on five benchmarks: AQuA (Ling et al., 2017), GPQA (Rein et al., 2024), MATH-500 (Lightman et al., 2023), GSM8K (Cobbe et al., 2021), and SVAMP (Patel et al., 2021). Following standard practice, evaluation is based on the accuracy of the final answer, independent of the CoT

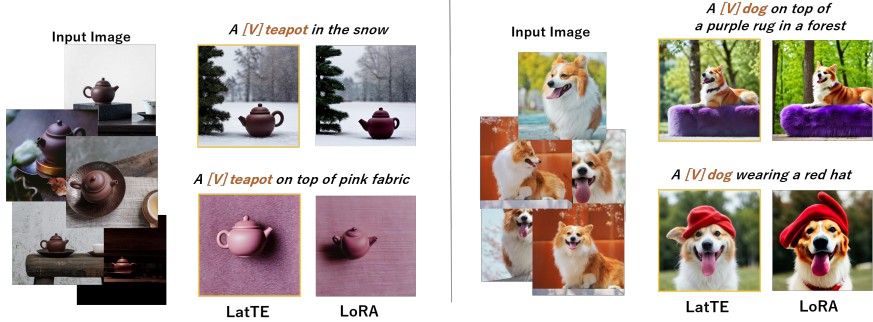

Figure 3: Subject-driven generation of LoRA and LatTE. All examples share the same seed for the two methods.

| Inference type | Split | Base (=merged LoRA) | Unmerged LoRA | LatTE-m | LatTE-w |
|---|---|---|---|---|---|
| Single example | All | 0.236 | 0.294 | 0.243 | 0.264 |
| | Attention | 0.126 | 0.168 | 0.133 | 0.154 |
| | FFN | 0.051 | 0.066 | 0.053 | 0.053 |
| | Embedding | 0.059 | 0.060 | 0.057 | 0.057 |
| 8 batch (1 task) | All | 0.229 | 0.285 | 0.238 | 0.254 |
| | Attention | 0.138 | 0.174 | 0.145 | 0.160 |
| | FFN | 0.047 | 0.062 | 0.047 | 0.047 |
| | Embedding | 0.044 | 0.049 | 0.046 | 0.047 |
| 8 batch (4 task) | All | - | - | 0.247 | 0.251 |
| | Attention | - | - | 0.153 | 0.155 |
| | FFN | - | - | 0.049 | 0.050 |
| | Embedding | - | - | 0.045 | 0.046 |

Table 4: Time per output token (in milliseconds) for Base, LoRA, and LatTE for Qwen2.5-7B (adapters on half of layers) with H100 GPU, context length 10k, averaged over 10 runs. The time is splited into attention, FFN, and embedding.

content. Table 2 presents results for Llama-3.2-1B/3B and Qwen2.5-1.5B/3B. Again, LatTE presented overall good results compared to the baselines, demonstrating its applicability to reasoning.

### 4.3 TEXT-TO-IMAGE GENERATION

We now turn to latent diffusion models and demonstrate that LatTE can also be applied to image generation models. Specifically, we use DreamBooth (Ruiz et al., 2023) dataset on Stable Diffusion v1.5 and test LoRA and LatTE's adaptation to subject-driven generation. We follow the default settings of the Huggingface Diffusers library for LoRA finetuning and applied LatTE to the identical positions, which are the attention blocks of the U-net (Ronneberger et al., 2015).

The qualitative results are shown in Figure 3. One observes that LoRA and LatTE both show effectiveness in subject-driven generation. However, LatTE's enlarged embedding cannot pass the base-model convolution layer without engaging with $f_{out}$. Therefore, multiple expansion and compression should be done for the forward pass, actually introducing overhead to inference. One can skip-connect the extra dimension after the convolutional layer to overcome this, however, the effectiveness of this strategy is yet to be explored.

## 5 DISCUSSION

**Effect of $f_{in}$ and $f_{out}$.** While LatTE does not introduce numerical hyperparameters beyond the rank $r$, it does involve several architectural design choices. As discussed in Section 3, these include: (1) the implementation of multi-head attention, (2) the initial expansion function $f_{in}(x)$, and (3) the final compression function $f_{out}([y, \bar{y}])$.

In addition to our default settings for $f_{in}$ and $f_{out}$, we consider several plausible alternatives. For the expansion function $f_{in}$, we experiment with $f_{in}(x) = [x; \mathbf{0}_r]$, where the auxiliary embedding is an $r$-dimensional zero vector $\mathbf{0}_r$. For the compression function $f_{out}$, we test discarding the auxiliary output, $f_{out}([y, \bar{y}]) = y$. $f_{in}(x) = [x; \text{pool}(x)]$, where the auxiliary embedding is derived via pooling, and $f_{out}([y, \bar{y}]) = y + \text{repeat}(\bar{y})$ was also considered but showed subobtimal performance. Table 3 reports the impact of these alternatives on commonsense QA performance.

**Inference Efficiency.** We also evaluate the inference-time efficiency of LatTE by measuring the time-per-output-token (TPOT). Figure 4 plots TPOT with the

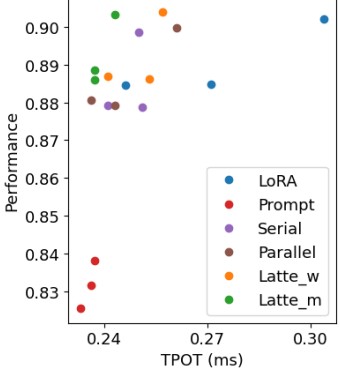

Figure 4: QA task score and inference speed on Qwen2.5-7B.

QA score with 7, 14, and 28 layers of adapters applied to Qwen2.5-7B model. We measured the inference time of 100 token generation with 10k context length, and averaged over 10 trials. OFT and BOFT results are not included as they were an order of magnitude slower than the others. One observes that the LatTE methods achieve suitable balance between performance and speed.

We present the TPOT results in Table 4, for Qwen2.5-7B, adapters on half layers. The context length is 10k and the results are averaged over 10 runs. This demonstrates LatTE's key advantage: constant-time inference regardless of the number of adapters and enabling multi-task batch inference, making it valuable for personalized and multi-domain deployment scenarios where merged LoRA's linear scaling becomes prohibitive and cannot be deployed in multi-task batch scenarios without constant load-unloading.

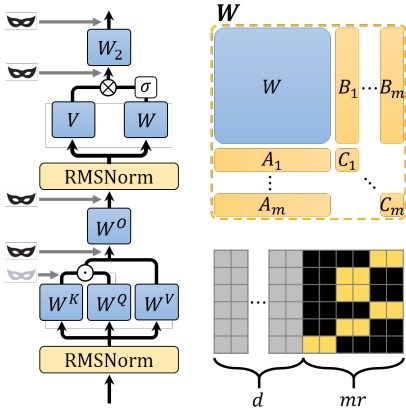

The inference efficiency also includes flexibility in intra-batch multi-adapter. While it is generally challenging to use multi-adapter within a batch, LatTE naturally supports such inference with the help of masking. When applying $m$ LatTEs, the inference batch will have $mr$ auxiliary embeddings. The task-dependent mask can be generated and applied throughout the forward pass to ensure the correct results. For example, the mask shown in Figure 5 is for a batch with tasks [2, 1, 1, 2, 1, 0]. This is applied in four positions per Transformer block as indicated with the mask symbol. Another mask is required in $(W_h^K x)^\top W_h^Q x$

Figure 5: Masking (black squares) in intra-batch multi-adapter scenario. The mask symbol indicates where masking should be applied for Transformers.

but can be merged with the rotary embeddings (Su et al., 2024). With masking, LatTE enables efficient intra-batch multi-task inference within a shared model backbone – highlighting its scalability for real-world multi-task applications.

## 6 CONCLUSION

We presented LatTE, a novel PEFT method that leverages auxiliary latent embeddings to achieve fast, scalable, and composable adaptation of large pretrained models. LatTE operates through a simple expansion of the embedding space, enabling task-specific adaptation via a single matrix multiplication per weight while keeping the base model frozen. Through extensive experiments we demonstrated that LatTE matches or exceeds the performance of strong PEFT baselines while offering significant improvements in inference efficiency. Beyond its empirical benefits, LatTE introduces a flexible design space for auxiliary embedding interactions. We anticipate that future work will build upon the core ideas of LatTE, and hope it serves as a foundation for further innovation.

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

| Model | Fintuning | Accuracy | | | | | | | | |
|-------|-----------|------|-------|------|------|--------|-------|-------|-------|------|
| | | OBQA | BoolQ | PIQA | SIQA | Hella. | Wino. | ARC-e | ARC-c | ave. |
| $r = 8$ | Prompt | 85.60 | 85.55 | 76.18 | 76.01 | 78.65 | 67.68 | 95.64 | 88.44 | 81.72 |
| | Serial | 89.95 | 88.43 | 88.07 | 79.18 | 91.89 | 80.82 | 95.62 | 89.40 | 87.92 |
| | Parallel | 90.44 | 88.70 | 88.66 | 78.36 | 92.27 | 82.04 | 95.35 | 88.72 | 88.07 |
| | OFT | 90.30 | 88.65 | 87.46 | 78.21 | 90.40 | 80.51 | 96.40 | 90.38 | 87.79 |
| | BOFT | 90.00 | 88.43 | 87.60 | 78.25 | 91.13 | 80.62 | 96.35 | 90.02 | 87.80 |
| | LoRA | 90.90 | 89.27 | 88.97 | 78.45 | 92.42 | 82.87 | 95.81 | 88.91 | 88.45 |
| | **LatTE-m** | 92.10 | 89.21 | 88.52 | 78.80 | 92.67 | 82.20 | 95.56 | 89.80 | 88.61 |
| | **LatTE-w** | 91.20 | 89.44 | 88.74 | 79.72 | 92.56 | 82.46 | 95.95 | 89.44 | **88.69** |
| $r = 16$ | Prompt | 87.95 | 85.24 | 86.60 | 76.67 | 76.44 | 66.75 | 96.39 | 89.23 | 83.16 |
| | Serial | 89.64 | 87.98 | 87.91 | 79.30 | 91.66 | 81.14 | 95.79 | 89.57 | 87.87 |
| | Parallel | 90.90 | 88.32 | 87.85 | 79.11 | 91.94 | 80.27 | 95.63 | 89.29 | 87.92 |
| | OFT | 90.35 | 88.49 | 88.40 | 79.49 | 91.89 | 82.20 | 96.32 | 89.78 | 88.36 |
| | BOFT | 90.00 | 88.43 | 87.60 | 78.25 | 91.13 | 80.62 | 96.35 | 90.02 | 87.80 |
| | LoRA | 91.10 | 89.05 | 88.29 | 78.68 | 92.67 | 83.09 | 95.57 | 89.38 | 88.48 |
| | **LatTE-m** | 91.70 | 89.34 | 88.79 | 79.45 | 92.20 | 83.80 | 95.91 | 89.69 | **88.86** |
| | **LatTE-w** | 91.20 | 89.26 | 88.55 | 78.90 | 91.94 | 83.19 | 95.76 | 90.17 | 88.62 |
| $r = 32$ | Prompt | 85.80 | 83.75 | 86.28 | 76.26 | 70.95 | 62.49 | 96.03 | 88.82 | 81.30 |
| | Serial | 89.50 | 88.29 | 87.99 | 79.35 | 91.64 | 80.80 | 95.80 | 89.68 | 87.88 |
| | Parallel | 90.70 | 88.52 | 88.15 | 79.55 | 92.00 | 80.60 | 95.75 | 89.31 | 88.07 |
| | OFT | 90.55 | 89.20 | 88.48 | 79.26 | 92.20 | 83.28 | 96.26 | 89.93 | 88.65 |
| | BOFT | 91.50 | 89.32 | 88.23 | 79.08 | 92.37 | 83.11 | 96.37 | 89.68 | 88.71 |
| | LoRA | 91.64 | 89.14 | 89.01 | 78.43 | 92.64 | 83.52 | 95.40 | 88.78 | 88.57 |
| | **LatTE-m** | 92.04 | 89.33 | 88.69 | 80.12 | 92.21 | 82.32 | 96.30 | 90.27 | 88.91 |
| | **LatTE-w** | 92.44 | 89.07 | 88.69 | 79.93 | 92.29 | 82.64 | 96.02 | 90.34 | **88.93** |

Table 5: Rank sensitivity analysis on Qwen2.5-7B commonsense QA with adapters in half layers. Both LatTE and LoRA show consistent improvement with increased rank, with LatTE maintaining competitive performance across all settings.

# A  PROOF OF THEOREM

**Theorem 2.** Any attention matrix produced by LoRA can be represented as an attention matrix from LatTE, given that $\bar{x}$ is a linear transform of $x$.

**Proof.**

i) Attention matrix of LoRA:

$$x^\top \left((W_k + \Delta W_k)^\top (W_Q + \Delta W_Q)\right) x$$
$$= x^\top \left((W_k + b_k^\top a_k^\top)^\top (W_Q + a_Q b_Q)\right) x \tag{4}$$

ii) Attention matrix of LatTE:

$$\begin{bmatrix} x^\top & \bar{x}^\top \end{bmatrix} \begin{bmatrix} W_K^\top & A_K^\top \\ B_K^\top & C_K^\top \end{bmatrix} \begin{bmatrix} W_Q & B_Q \\ A_Q & C_Q \end{bmatrix} \begin{bmatrix} x \\ \bar{x} \end{bmatrix}$$
$$= x^\top (W_K^\top W_Q + B_K^\top B_Q) x + x^\top (W_K^\top A_Q + B_K^\top C_Q)\bar{x}$$
$$+ \bar{x}^\top (A_K^\top W_Q + C_K^\top B_Q) x + \bar{x}^\top (A_K^\top A_Q + C_K^\top C_Q)\bar{x} \tag{5}$$

We claim that any LoRA attention can be expressed by LatTE attention. Let $A_Q = a_Q$, $A_K = a_K$, $B_Q = a_K^\top W_Q$, $C_Q = a_K^\top a_Q$, $B_K$, $C_K$ satisfies $B_K + C_K b_Q = b_K - b_Q$, and $\bar{x} = b_Q x$. Note that the solution for $B_K + C_K b_Q = b_K - b_Q$ always exists since $B_K = b_K$ and $C_K = -1_r$ (identity matrix) satisfies the condition.

| Model | Fintuning | Accuracy | | | | | | | | | Degradation |
|---|---|---|---|---|---|---|---|---|---|---|---|
| | | OBQA | BoolQ | PIQA | SIQA | Hella. | Wino. | ARC-e | ARC-c | ave. | |
| Qwen2.5-14B (BF16) | LoRA | 94.20 | 90.30 | 91.07 | 81.41 | 93.85 | 87.04 | 97.61 | 93.28 | 91.09 | - |
| | **LatTE-m** | 94.10 | 90.31 | 91.01 | 80.90 | 94.07 | 87.12 | 97.63 | 93.58 | 91.09 | - |
| | **LatTE-w** | 94.10 | 90.47 | 91.57 | 81.12 | 94.09 | 87.25 | 97.54 | 93.75 | **91.24** | - |
| Qwen2.5-14B (INT8) | LoRA | 93.76 | 90.22 | 90.92 | 80.67 | 93.51 | 86.86 | 97.50 | 93.26 | 90.84 | 0.27% |
| | **LatTE-m** | 94.24 | 90.41 | 91.02 | 81.01 | 93.93 | 86.54 | 97.47 | 93.47 | 91.01 | 0.09% |
| | **LatTE-w** | 94.04 | 90.24 | 91.51 | 80.67 | 93.86 | 87.39 | 97.48 | 93.36 | **91.07** | 0.19% |

Table 6: Accuracy results on the commonsense QA benchmark for Qwen2.5-14B model, with BF16 and INT8 precision. Adapters are applied to half of the layers. LatTE shows robust performance retention under quantization, demonstrating compatibility with standard inference acceleration techniques.

Then the LatTE attention becomes:

$$
\begin{aligned}
x^\top (W_K^\top & W_Q + B_K^\top B_Q + W_K^\top a_Q b_Q + B_K^\top C_Q b_Q \\
& + b_Q^\top a_K^\top W_Q + b_Q^\top C_K^\top B_Q + b_Q^\top a_K^\top a_Q b_Q + b_Q^\top C_K^\top C_Q b_Q)x \\
= x^\top (W_K^\top & + b_K^\top a_K^\top)(W_Q + a_Q b_Q)x \\
& + x^\top ((b_Q - b_K)^\top a_K^\top W_Q + (b_Q - b_K)^\top a_K^\top a_Q b_Q \\
& \quad + (B_K + C_K b_Q)^\top B_Q + (B_K + C_K b_Q)^\top C_Q b_Q)x
\end{aligned}
\tag{6}
$$

The second term identically vanishes with $C_Q = a_K^\top a_Q$, $B_Q = a_K^\top W_Q$ and $B_K + C_K b_Q = b_K - b_Q$. This proves any LoRA attention matrix can be expressev as LatTE attention matrix assuming $\bar{x}$ is a linear transform of $x$.

# B ADDITIONAL RESULTS

Here we provide additional experimental results.

## B.1 ABLATION ON EMBEDDING SIZE

We show the ablation study on embedding size ($r$) on Qwen2.5-7B for commonsense QA benchmark in Table 5. The scaling behavior of LatTE for $r$ is similar to that of LoRA, as expected. The consistent relative performance across ranks indicates LatTE and LoRA have similar expressivity – increasing rank benefits both methods equally, showing no fundamental expressivity gap.

## B.2 LARGER MODELS AND EFFECT OF QUANTIZATION

To address the issue of scalability, we present results for larger model size. Table 6 shows results for Qwen2.5-14B for commonsense QA benchmark, with adapters in half of the layers. LatTE models showed competitive accuracy compared to LoRA, consistent with the smaller models, with LatTE-w achieving the highest average accuracy. While this is not as large a model as tens or hundreds of billions of parameters, we experimented on a range of parameters (1-14B) which showed consistent effectiveness. Together with the theoretical analysis, we believe that the effectiveness of LatTE will hold for substantially larger models.

We also consider the effect of quantization, and quantize the weights to INT8 for the same model. The results showed that LatTE was equally effective even for quantized models, and the accuracy degradation from quantization is of similar level to LoRA.

## B.3 EPOCH-WISE PERFORMANCE

For the Commonsense QA task, we show the performance of each epoch in Figure 6

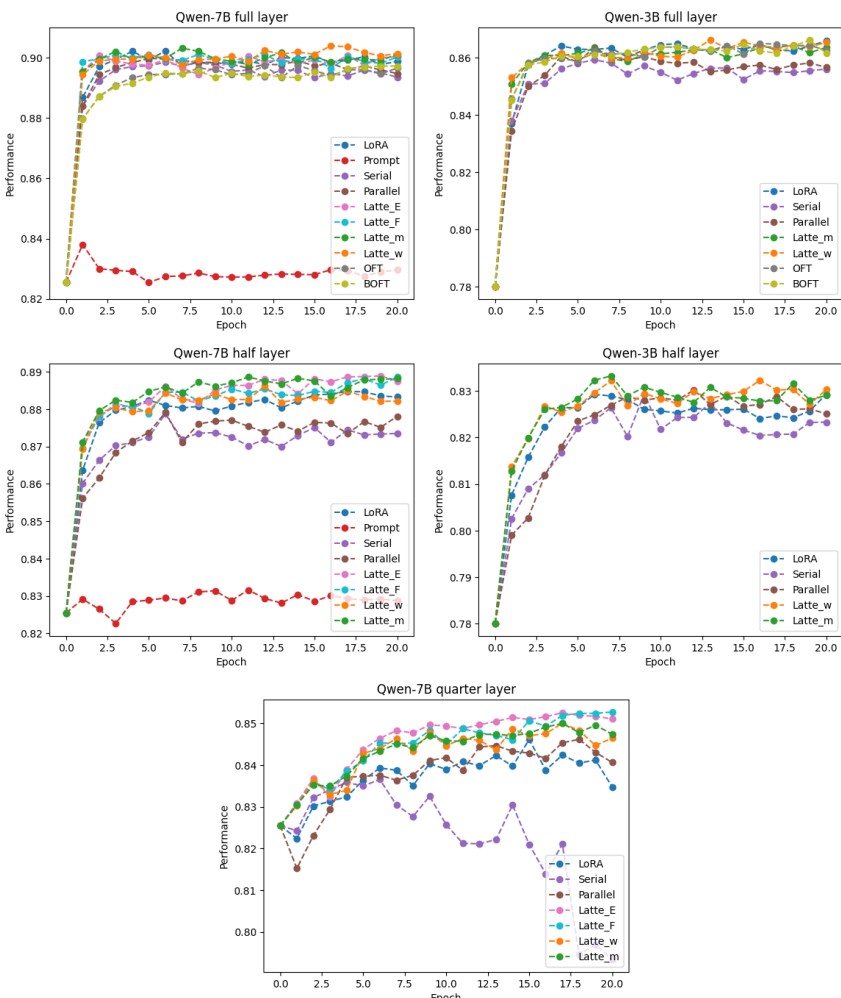

Figure 6: Epoch-wise performance for QA task.

### B.4 MORE RESULTS ON TEXT-TO-IMAGE GENERATION

More qualitative results for the subject-driven generation is shown in Figure 7.

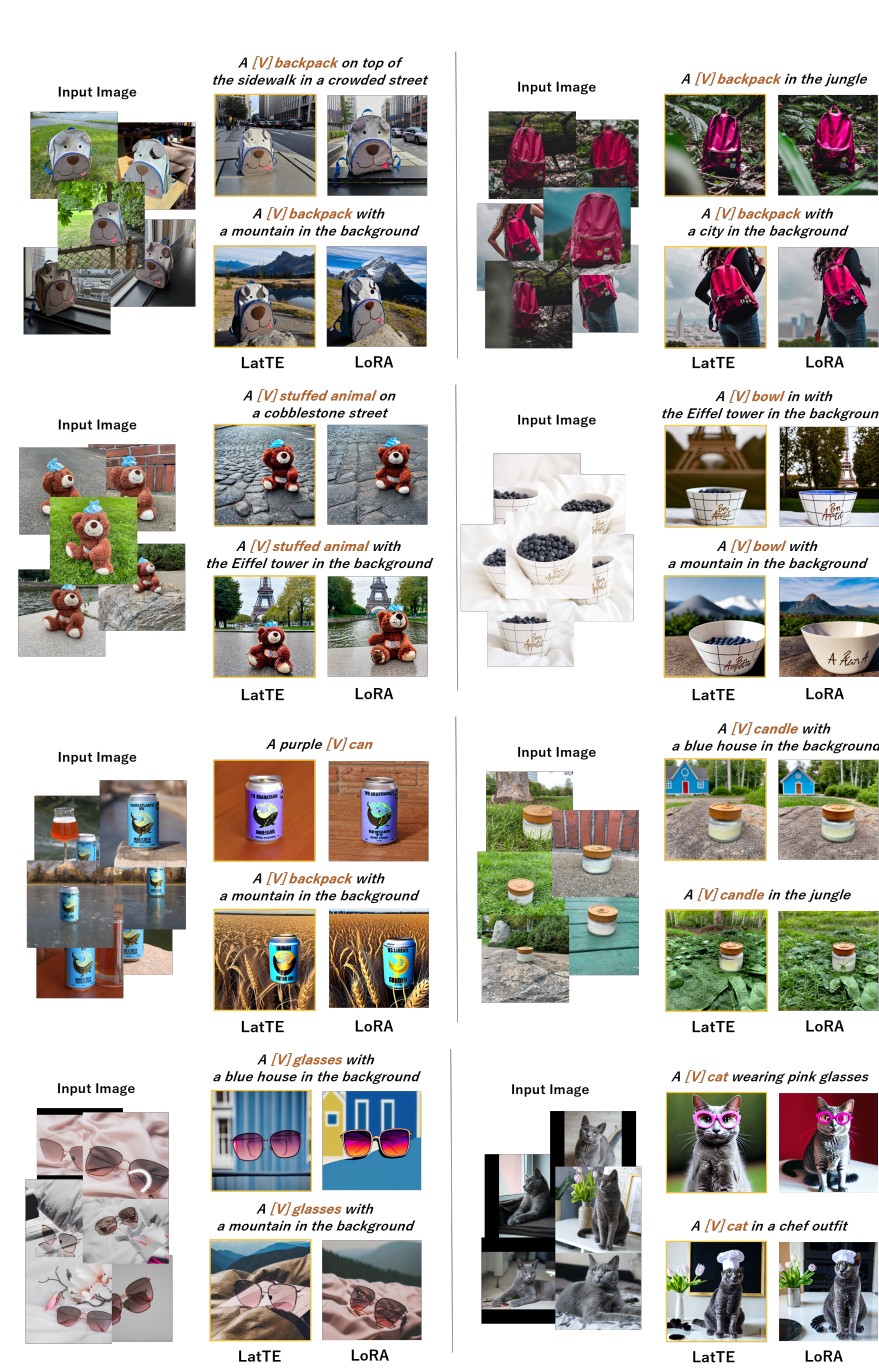

Figure 7: Additional results for subject-driven generation of LoRA and LatTE.

