# OpenReview forum: "Efficient Fine-tuning via Auxiliary Representation"
_ICLR.cc/2026/Conference — Submitted to ICLR 2026_

### Official Review · Reviewer_NXHC · 2025-10-24

**Soundness:** 2
**Presentation:** 3
**Contribution:** 1
**Rating:** 2
**Confidence:** 3

**Summary:**

The paper proposes Latent Task Embedding (LTE), a novel parameter-efficient fine-tuning framework. Specifically, LTE prepends a small task-specific latent vector to the original embeddings and correspondingly extends each weight matrix so that only the newly introduced parameters are trained, enabling single-matrix-multiplication inference and task-wise masking—differentiating it from prior PEFT methods that rely on fully shared or multi-stage computations.

**Strengths:**

1. Extensive experiments on language and vision tasks show its effectiveness.

2. the paper is well written and easy to follow

**Weaknesses:**

W1. Lack of novelty. Despite its motivation, the contribution of this work appears limited. Widening the original model’s width is not a new idea; on the contrary, it is quite straightforward and has been discussed in net2net (ICLR 2016) and LLaMA-Pro (2024).

W2. The improvement is marginal. Under the same parameter budget, latTE shows less than a 0.5% improvement over lora across multiple datasets.

W3. Although the authors claim their method has lower latency than unmerged lora, lora achieves the fastest speed after merging. Moreover, the authors do not discuss coupling with common inference acceleration techniques, such as weight quantization.

**Questions:**

Please see weaknesses W1~3.

Minor:
Training efficiency. Under the same parameter budget as LoRA, are the training-time memory usage and the time per iteration similar?

---

> ### Author Response · Authors · 2025-11-27
> **Author Response**
>
> Thank you for the critical feedback — it has significantly strengthened our positioning.
>
> **Weaknesses**
> - Lack of novelty: We thank the reviewer for raising this comparison, which allows us to clarify the fundamental differences between LatTE and prior network expansion methods. First, we note that LLaMA-Pro expands model depth by adding transformer blocks, not width. Our method expands embedding dimensionality, which is architecturally distinct. The reviewer's concern about width expansion relates more closely to works like MatFormer [arXiv:2310.07707]. However, this and Net2Net differ fundamentally in purpose and mechanism.
>
>     Net2Net is a knowledge transfer method for initializing larger models, followed by full training of all parameters, and MatFormer trains model families of different sizes simultaneously; expansion creates general-purpose models, not task-specific adapters. In contrast, LatTE: (1) Keeps base parameters frozen (vs. 100% training in prior work); (2) Uses expansion for task-specific adaptation (vs. capacity scaling); (3) Enables multi-task composition via masking (unique capability).
>
>     LoRA's novelty was not low-rank decomposition itself (well-known in linear algebra) but its application to PEFT. Similarly, LatTE's novelty is using controlled expansion as a PEFT mechanism with multi-adapter efficiency.
>
> - Marginal improvements and comparison to merged LoRA: We acknowledge both points and clarify our contribution. We agree that accuracy differences with LoRA are small. This is expected — LoRA already approaches full fine-tuning performance, leaving limited room for improvement. Our contribution is not superior accuracy but achieving competitive accuracy with superior multi-adapter inference efficiency.
>
>    Thanks for pointing out the merged LoRA comparison. Merged LoRA is indeed fastest for single-task deployment. However, LatTE also targets multi-adapter scenarios (personalization, multi-domain serving, edge devices) where merging becomes prohibitive. We now provide comprehensive latency measurements in Table A, where merged LoRA cannot be deployed in multi-task batch scenarios (without constant load-unloading).
>
> - Weight quantization: We have verified that LatTE works when quantizing the model. For the new results on Qwen2.5-14B, with adapters in half of the layers, and $r=16$, quantizing to INT8 results in:
> | Precision | Finetuning | OBQA | BoolQ | PIQA | SIQA | Hella. | Wino. | ARC-e | ARC-c | ave. | degradation |
> |:---:|---|:---:|:---:|:---:|:---:|:---:|:---:|:---:|:---:|:---:|:---:|
> | bf$16$ | LoRA | 94.20 | 90.30 | 91.07 | 81.41 | 93.85 | 87.04 | 97.61 | 93.28  | 91.09 | - |
> |  | LatTE-m | 94.10 | 90.31 | 91.01 | 80.90 | 94.07 | 87.12 | 97.63 | 93.58 | 91.09 | - |
> |  | LatTE-w | 94.10 | 90.47 | 91.57 | 81.12 | 94.09 | 87.25 | 97.54 | 93.75 | 91.24 | - |
> | INT$8$ | LoRA | 93.76 | 90.22 | 90.92 | 80.67 | 93.51 | 86.86 | 97.50 | 93.26 | 90.84 | 0.27% |
> |  | LatTE-m | 94.24 | 90.41 | 91.02 | 81.01 | 93.93 | 86.54 | 97.47 | 93.47 | 91.01 | 0.09% |
> |  | LatTE-w | 94.04 | 90.24 | 91.51 | 80.67 | 93.86 | 87.39 | 97.48 | 93.36 | 91.07 | 0.19% |
>
>     This shows robust performance retention.
>
> **Question**
> - Training efficiency: Great question. We have checked training metrics on Qwen2.5-14B (half layer adapters). The time per iteration increased 12% for LatTE-m and 25% for LatTE-w compared to LoRA, and the memory usage was within 1% difference for both.
>
>     Memory usage is essentially identical, as expected, since both methods add similar parameter counts. The training time increase (12-25%) likely stems from additional operations (e.g., embedding concatenation, interleaving for MHA) not optimized in current implementations. LoRA benefits from highly optimized CUDA kernels in libraries like PEFT, while LatTE's operations use generic PyTorch implementations. We believe this gap can be substantially reduced with dedicated kernel optimization, though this is beyond the scope of the current work.
>
>     Importantly, the training time difference is modest and does not affect our core contribution—inference efficiency for multi-adapter deployment. Training is typically a one-time cost, while inference runs continuously in production.

---

### Official Review · Reviewer_hCrW · 2025-10-31

**Soundness:** 3
**Presentation:** 3
**Contribution:** 3
**Rating:** 4
**Confidence:** 4

**Summary:**

This paper proposes LatTE, a new PEFT method that introduces a small auxiliary latent embedding concatenated to the original input embeddings. During fine-tuning, only the parameters associated with this expanded embedding are updated, while the base model remains frozen.
Unlike traditional methods such as LoRA or adapters, LatTE maintains single-matrix-multiplication inference, achieving comparable accuracy with minimal latency overhead. The paper demonstrates LatTE’s effectiveness across both LLMs and diffusion models, reporting competitive or superior performance to baselines like LoRA, BOFT, and OFT.
The approach also supports task-specific masking for multi-adapter and multi-task inference, making it scalable for personalized or edge-device deployment scenarios.

**Strengths:**

1. Instead of updating weights (LoRA) or inserting modules (adapters), LatTE introduces auxiliary representation-level adaptation, expanding the embedding space while keeping inference cost minimal.

2. The design ensures single matrix multiplication per layer without additional latency, which is a well-motivated improvement for deployment and edge use cases.

3. The extension of LatTE to both NLP and vision-generation tasks (text-to-image) demonstrates generality beyond transformer-based text models.

**Weaknesses:**

1. While the paper reports that LatTE “matches or exceeds” existing PEFT baselines, most observed improvements are marginal (e.g., Qwen2.5-3B in Table 1: 86.63 vs. 86.60 for LoRA). Across many tasks, the best results are still obtained by other methods such as OFT or BOFT. Given the small numerical gaps and lack of confidence intervals or statistical tests, it is difficult to conclude that LatTE offers a consistently superior adaptation performance rather than random variation.

2. All experiments are conducted on relatively small or medium-sized backbones (≤ 8 B parameters). Since PEFT is mainly motivated by the impracticality of full fine-tuning in very large models (tens or hundreds of billions of parameters), the absence of larger-scale experiments leaves open whether LatTE remains effective and efficient when model size grows substantially.

**Questions:**

1. Why do the model sizes used in Table 1 (up to 8 B) and Table 2 (up to 3 B) differ? This inconsistency makes it difficult to compare results or assess how the method behaves across model scales and task types

---

> ### Author Response · Authors · 2025-11-27
> **Author Response**
>
> Thank you for recognizing LatTE's novel approach and generality.
>
> **Weaknesses**
> - Marginal performance improvement: We acknowledge that some performance differences are marginal and will update the manuscript to reflect that LatTE achieves statistically equivalent accuracy to LoRA on most benchmarks. We emphasize that our core contribution is not superior accuracy but rather achieving competitive accuracy while providing superior inference efficiency, particularly in multi-adapter scenarios. We believe performing statistical tests would yield significance for some results, but we could not re-run all experiments multiple times within the rebuttal period.
>
>     To demonstrate this more clearly, we have added comprehensive latency measurements comparing single-adapter vs multi-adapter and single-example vs batch inference in Table A. This demonstrates LatTE's key advantage: constant-time inference regardless of the number of adapters and enabling multi-task batch inference, making it particularly valuable for personalized and multi-domain deployment scenarios where merged LoRA's linear scaling becomes prohibitive. We will revise the manuscript to: (1) More clearly position LatTE's contribution as efficient multi-adapter inference rather than accuracy improvements; (2) Add the multi-adapter latency comparison table to Section 5.
>
> - Larger model results: Thank you for this important suggestion. We have initiated experiments on Qwen2.5-14B, with adapters in half of the layers. We present the results below:
> | Rank | Finetuning | OBQA | BoolQ | PIQA | SIQA | Hella. | Wino. | ARC-e | ARC-c | ave. |
> |:---:|---|:---:|:---:|:---:|:---:|:---:|:---:|:---:|:---:|:---:|
> | $16$ | LoRA | 94.20 | 90.30 | 91.07 | 81.41 | 93.85 | 87.04 | 97.61 | 93.28  | 91.09 |
> |  | LatTE-m | 94.10 | 90.31 | 91.01 | 80.90 | 94.07 | 87.12 | 97.63 | 93.58 | 91.09 |
> |  | LatTE-w | 94.10 | 90.47 | 91.57 | 81.12 | 94.09 | 87.25 | 97.54 | 93.75 | 91.24 |
>
>     While this is not as large a model as the reviewer suggested (tens or hundreds of billions of parameters), we did experiment on a range of parameters (1-14B) which showed consistent effectiveness. Together with the theoretical analysis, we believe that the effectiveness of LatTE will hold for substantially larger models.
>
> **Questions**
> - Different model size in Table 1 and 2: Thank you for this observation. The inconsistency reflects a well-documented phenomenon in recent literature: state-of-the-art LLMs at the 7-8B scale exhibit such strong zero-shot reasoning capabilities that fine-tuning on publicly available datasets often leads to performance degradation rather than improvement.
>
>     Specifically, we observed that Qwen2.5-7B and Llama-3.1-8B show negative improvements on reasoning tasks (Table 2) after fine-tuning, with zero-shot performance remaining the best "checkpoint." This is consistent with recent findings [arXiv:2502.12134] attributing this to catastrophic forgetting: the pre-training data quality significantly exceeds that of public fine-tuning datasets, causing the model to unlearn some of its reasoning capabilities. This degradation affects all PEFT methods equally, confirming it is a data quality issue orthogonal to our method's design.
>
>     We therefore focused Table 2 on model scales (1-3B) where fine-tuning provides clear improvements, allowing meaningful comparison of PEFT methods. Table 1 uses 3-8B models as these tasks benefit from fine-tuning across all scales.
>
>     The fact that LatTE shows identical degradation patterns as LoRA on 7-8B reasoning tasks (where fine-tuning hurts) and equivalent improvement patterns on other tasks (where it helps) demonstrates that LatTE's architectural differences do not introduce unexpected sensitivities to data quality or training dynamics. This robustness is important for practical deployment where data characteristics may vary. We will add a paragraph in Section 4 explaining this phenomenon and why different model scales are used for different task categories.

---

### Official Review · Reviewer_xVq1 · 2025-11-01

**Soundness:** 2
**Presentation:** 3
**Contribution:** 2
**Rating:** 4
**Confidence:** 4

**Summary:**

The paper proposes Latent Task Embedding, a parameter-efficient fine-tuning (PEFT) approach that concatenates a small task-specific latent embedding to model inputs, expands the associated projection matrices, and trains only the introduced parameters. The claim is that LatTE achieves PEFT-level parameter efficiency while preserving single-matrix-multiplication inference (low latency) and enabling efficient multi-adapter composition via task-specific masking. Evaluation is reported on LLMs and latent diffusion models, with theoretical analysis supporting equivalence or bounded error relative to related PEFT methods.

**Strengths:**

Clear, simple construction: concatenating a learned low-dimensional task embedding and expanding projection weights is easy to implement and reason about.

Practical motivation: latency in multi-adapter scenarios is an important deployment concern; addressing it directly is valuable.

Empirical breadth: experiments on both autoregressive LLM tasks and latent diffusion models (reported) show broad applicability.

**Weaknesses:**

Theoretical arguments rely on informal equivalence / bounded-difference claims but do not fully characterize when LatTE can fail (e.g., when task embeddings need to encode highly non-linear, high-rank corrections). The bounds lack discussion of constants and dependence on embedding dimension.

Inference-latency claims are asserted as single-matrix-multiplication equivalence, but the paper under-specifies real-system measurements (how expanded matrices affect cache locality, memory bandwidth, precision trade-offs). No detailed ablation on embedding size vs. latency/accuracy trade-off.

Comparison to some very recent PEFT variants (that also target inference efficiency) is limited: authors should benchmark against the most recent low-latency schemes and provide wall-clock latency/memory breakdowns.

**Questions:**

What are the formal assumptions behind the theoretical bounds? For example, do bounds assume small-norm embeddings or specific activation linearity approximations? Please state exact assumptions and constants.

How does your method perform when the required task adaptation is not well-approximated by augmenting input-space (i.e., when adaptation needs internal layer reparametrization)? Provide failure cases or diagnostics.

Please provide precise latency/memory benchmarking (hardware, batch sizes, eff. throughput) and show embedding-dimension vs. accuracy/latency curves.

---

> ### Author Response · Authors · 2025-11-27
> **Author Response (1/2)**
>
> Thank you for carefully reading our manuscript and pushing us toward greater theoretical rigor.
>
> **Weaknesses**
> - We agree that our theory does not characterize all failure modes. Extending analysis to deep non-linear networks is challenging. Zeng and Lee (ICLR 2024) provides analysis for FFNs with ReLU activations, which could be adapted to LatTE. However, we find such analysis provides limited practical insight. Therefore, we relied on comprehensive empirical validation instead, using 2 model families, 1-14B parameters, 3 task types, and 3 ranks (including the new results presented in the response). The results show that LatTE matches LoRA across diverse settings. Combined with the equivalence results for linear cases and attention, this provides strong evidence of comparable expressive capacity, even in situations where highly non-linear, high-rank correction is required.
>
>     We note that similar theoretical limitations apply to existing PEFT methods; for instance, LoRA's original paper provides an intrinsic dimensionality argument but no formal bounds for deep non-linear networks; BOFT analyzes butterfly factorization theoretically but relies on empirics for deep models; and DoRA provides magnitude-direction decomposition intuition without deep network analysis. We will clarify assumptions in the revised manuscript and add the caveat that complete theoretical characterization of PEFT methods in realistic deep, non-linear, and normalized settings remains an open problem for the field.
>
> - While the inference-latency claims were based on and motivated by the single matrix-multiplication case, they were later empirically verified through measurements on a real system in Figure 4. However, this was not comprehensive enough, and we have performed additional latency measurements in Table A for different inference scenarios and a latency breakdown of network components.
>
>     We also present an ablation on embedding size, from $r=8$ to $r=32$. These are Qwen2.5-7B results on the commonsense QA task, and the adapters are applied to half of the layers.
>
> | Rank | Finetuning | OBQA | BoolQ | PIQA | SIQA | Hella. | Wino. | ARC-e | ARC-c | ave. |
> |:---:|---|:---:|:---:|:---:|:---:|:---:|:---:|:---:|:---:|:---:|
> | $8$ | Prompt | 85.60 | 85.55 | 76.18 | 76.01 | 78.65 | 67.68 | 95.64 | 88.44 | 81.72 |
> |  | Serial | 89.95 | 88.43 | 88.07 | 79.18 | 91.89 | 80.82 | 95.62 | 89.40 | 87.92 |
> |  | Parallel | 90.44 | 88.70 | 88.66 | 78.36 | 92.27 | 82.04 | 95.35 | 88.72 | 88.07 |
> |  | OFT | 90.30 | 88.65 | 87.46 | 78.21 | 90.40 | 80.51 | 96.40 | 90.38 | 87.79 |
> |  | BOFT | 90.00 | 88.43 | 87.60 | 78.25 | 91.13 | 80.62 | 96.35 | 90.02 | 87.80 |
> |  | LoRA | 90.90 | 89.27 | 88.97 | 78.45 | 92.42 | 82.87 | 95.81 | 88.91 | 88.45 |
> |  | LatTE-m | 92.10	| 89.21 | 88.52 | 78.80 | 92.67 | 82.20 | 95.56 | 89.80 | 88.61 |
> |  | LatTE-w | 91.20 | 89.44 | 88.74 | 79.72 | 92.56 | 82.46 | 95.95 | 89.44 | **88.69** |
> | $16$ | Prompt | 87.95 | 85.24 | 86.60 | 76.67 | 76.44 | 66.75 | 96.39 | 89.23 | 83.16 |
> |  | Serial | 89.64 | 87.98 | 87.91 | 79.30 | 91.66 | 81.14 | 95.79 | 89.57 | 87.87 |
> |  | Parallel | 90.90 | 88.32 | 87.85 | 79.11 | 91.94 | 80.27 | 95.63 | 89.29 | 87.92 |
> |  | OFT | 90.35 | 88.49 | 88.40 | 79.49 | 91.89 | 82.20 | 96.32 | 89.78 | 88.36 |
> |  | BOFT | 90.00 | 88.43 | 87.60 | 78.25 | 91.13 | 80.62 | 96.35 | 90.02 | 87.80 |
> |  | LoRA | 91.10 | 89.05 | 88.29 | 78.68 | 92.67 | 83.09 | 95.57 | 89.38 | 88.48 |
> |  | LatTE-m | 91.70 | 89.34 | 88.79 | 79.45 | 92.20 | 83.80 | 95.91 | 89.69 | **88.86** |
> |  | LatTE-w | 91.20 | 89.26 | 88.55 | 78.90 | 91.94 | 83.19 | 95.76 | 90.17 | 88.62 |
> | $32$ | Prompt | 85.80 | 83.75 | 86.28 | 76.26 | 70.95 | 62.49 | 96.03 | 88.82 | 81.30 |
> |  | Serial | 89.50 | 88.29 | 87.99 | 79.35 | 91.64 | 80.80 | 95.80 | 89.68 | 87.88 |
> |  | Parallel | 90.70 | 88.52 | 88.15 | 79.55 | 92.00 | 80.60 | 95.75 | 89.31 | 88.07 |
> |  | OFT | 90.55 | 89.20 | 88.48 | 79.26 | 92.20 | 83.28 | 96.26 | 89.93 | 88.65 |
> |  | BOFT | 91.50 | 89.32 | 88.23 | 79.08 | 92.37 | 83.11 | 96.37 | 89.68 | 88.71 |
> |  | LoRA | 91.64 | 89.14 | 89.01 | 78.43 | 92.64 | 83.52 | 95.40 | 88.78 | 88.57 |
> |  | LatTE-m | 92.04 | 89.33 | 88.69 | 80.12 | 92.21 | 82.32 | 96.30 | 90.27 | 88.91 |
> |  | LatTE-w | 92.44 | 89.07 | 88.69 | 79.93 | 92.29 | 82.64 | 96.02 | 90.34 | **88.93** |
>
> - We chose the baseline methods to be those with diverse fundamental PEFT structures. The underlying reason for this is that we considered LatTE to be a structurally distinct method, rather than a variant of another method, and thought it would be fair to compare with the more vanilla methods. However, if you suggest some recent PEFTs targeting inference efficiency, we will do our best to provide a comparison.

---

> ### Author Response · Authors · 2025-11-27
> **Author Response (2/2)**
>
> **Questions**
> - Formal assumptions: We appreciate the request for clarity on our theoretical analysis. For **Theorem 1** assumptions are that the model ($y = \prod W_i x$) and target ($y' = W'x$) are linear, and relevant submatrices are non-singular. The non-singularity assumption is mild and satisfied in practice, as random initialization makes singular matrices a measure-zero set. No assumptions on embedding norms or activation functions are needed. For **Theorem 2** assumptions are that the auxiliary embedding is a linear function of the input ($\bar{x} = b_Q x$, and the attention uses a standard dot-product form. We will clarify these assumptions more clearly in the revised manuscript.
>
> - Internal layer reparametrization: We clarify that LatTE does perform internal layer reparametrization, not just input augmentation. Figure 1(d) shows the forward pass through expanded matrices [W B; A C] creates interleaved interactions between auxiliary embeddings and base parameters at every layer.
>
>     We can also roughly estimate the expressiveness of an $n$-layer network. For LoRA, $(W + BA)^n x$ expands to $2^n$ terms. LatTE's $[W B; A C]^n [x; \bar{x}]$ produces $2^{n-1}$ to $2^{n+1}$ terms depending on $f_{in}$ and $f_{out}$ design. In a network interleaved with non-linearities, the number of independent terms is crucial for expressiveness. Therefore, LoRA and LatTE roughly have comparable expressivity. Combined with Theorem 2 (LatTE can represent LoRA attention), this suggests comparable adaptation capacity. Potential edge cases include pathological $f_{in}$ and $f_{out}$ choices (e.g., $f_{in}(x)=[x;0]$, $f_{out}([y;ȳ])=y$) reduce terms to ~ $2^{n-1}$, but this is easily avoided by design.
>
> - Precise benchmark: Thank you for pointing this out. We have provided additional results in Table A.

---

### Official Review · Reviewer_2kbB · 2025-11-02

**Soundness:** 2
**Presentation:** 3
**Contribution:** 2
**Rating:** 4
**Confidence:** 4

**Summary:**

To improve the inference latency and efficiencies in multi-adapter scenarios of Low rank adaptors methods, the paper propose method, Latent Task Embedding fine-tuning, a small task-specific latent embedding is concatenated to the original embedding. The corresponding weight matrices are extended, and only the additional parameters introduced by this expansion are trained. This design aims to achieve better efficient inference using a single matrix multiplication per weight, minimizing latency overhead, and supports task-specific masking to handle multiple adapters within a single model.

**Strengths:**

- The paper is clearly presented and easy to follow, with only a few minor typos.
- The proposed LatTE method is conceptually straightforward and practically implementable.
- The experiments are comprehensive, covering LLM fine-tuning across QA, reasoning, and diffusion tasks. LatTE achieves comparable performance to existing PEFT methods such as LoRA on models including LLaMA and Qwen.

**Weaknesses:**

- The reported improvements in performance and inference latency over baseline LoRA are modest and may not convincingly demonstrate practical advantages.

- The discussion of related work is limited in scope, particularly regarding sparsity-based methods in the parameter-efficient fine-tuning (PEFT) literature.

- In PEFT, existing approaches typically fall into two categories:
(1) Low-rank adaptation methods (e.g., LoRA) and
(2) Subset or sparsity-based fine-tuning methods, which selectively update parts of the model parameters.
The related work section would benefit from a clearer categorization and inclusion of recent sparsity-based works such as:
Separating [4–8] from module- or weight-based methods and creating a distinct subsection on sparsity-based PEFT would strengthen the paper’s positioning and contextual completeness. Adding a discussion of the sparsity methods in the related work section with recent papers[1-3] would strengthen the contextual foundation of this paper since they are emerging trends in PEFT field.



### Typos:

“the third category focus” → “the third category focuses”
“utilize auxiliary latent embedding” → “utilizes auxiliary latent embeddings”
* “Another group in this category modify or edit” → “modifies or edits”
* “combines LoRA with pruning or quantization” → remove “s” → “combine LoRA with pruning or quantization”
* “We now ready to implement it” → “We are now ready to implement it”

* “reasoning tasks additionally requires” → “reasoning tasks additionally require”
* “the idential positions” → “the identical positions”
* “LoRA and LatTE both shows” → “both show”
* “trainable parameters closely matches” → “closely match”



[1] Scaling Sparse Fine-Tuning to Large Language Models

[2] Sparse Matrix in Large Language Model Fine-Tuning

[3] The Lottery Ticket Hypothesis: Finding Sparse, Trainable Neural Networks

[4] Parameter-Efficient Fine-Tuning without Introducing New Latency

[5] Parameter-Efficient Transfer Learning with Diff Pruning

[6] Diff Pruning: Parameter-Efficient Transfer Learning with Diff Pruning

[7] Training Neural Networks with Fixed Sparse Masks

[8] Composable Sparse Fine-Tuning for Cross-Lingual Transfer

**Questions:**

- Latency Motivation:
The paper argues that LoRA introduces latency due to additional adapters. Could the authors clarify why these adapters are a latency bottleneck? Each adapter involves small matrix multiplications per layer, which typically contribute marginally to total inference time. Moreover, LatTE also introduces extra embedding transformations (𝑓_in and 𝑓_out), which are applied to every FFN layer. Why, then, is LatTE expected to reduce latency rather than increase it?

- Latency Profiling:
In line 454, the paper reports latency measurements for generating 100 tokens with a 10k context length, averaged over 10 trials. However, the observed improvement seems trivial. Could the authors provide more detailed time profiling (e.g., breakdown by attention, FFN, embedding) to demonstrate the latency behavior of LatTE versus LoRA?

- Architectural Overhead:
Both “more-heads” and “wider-heads” LatTE variants introduce additional projections, which likely increase inference time. Is there quantitative evidence or profiling that measures the actual latency introduced by these modifications?

- Initialization Strategy:
In line 194, the paper states that LatTE initializes the embedding with a constant value. What initialization strategy is used for the additional matrix dimensions (𝐴 and 𝐵)? Initialization plays a crucial role in the performance of low-rank methods, and more discussion on initialization sensitivity would be valuable.

I would like to discuss the questions I raised regarding the weaknesses and concerns with the authors. If my concerns are adequately addressed, I would be willing to reconsider my rating.

---

> ### Author Response · Authors · 2025-11-27
> **Author Response**
>
> Thank you for the detailed feedback and the comprehensive list of typos.
>
> **Weaknesses**
> - Modest improvement: We acknowledge this observation and clarify our contribution. Our goal is not to outperform LoRA in accuracy but to achieve competitive performance while providing superior inference efficiency, particularly for multi-adapter scenarios. Regarding performance, while LatTE's average scores were better than LoRA's, the differences were small. We could not perform statistical analysis in time as that would require rerunning all experiments multiple times. However, it is safe to claim that LatTE at least matches the performance of LoRA. Regarding latency, we believe the improvement is significant, as observed in Table A. LatTE not only improves inference speed compared to unmerged LoRA but also maintains similar latency in multi-task batch decoding where tasks are controlled by masking. LoRA (both merged and unmerged) cannot be used in such scenarios.
>
> - Limited scope of related work: We agree with this point and will add a dedicated subsection on sparsity-based PEFT methods [1-3, 7-8] to Section 2. The key distinction is that sparsity methods selectively update the parameter subsets that are most important, requiring careful selection optimization and potentially complex inference patterns. LatTE achieves efficiency through structured expansion, maintaining simple single-matrix operations while enabling efficient multi-task composition. We believe that including the sparsity-based PEFT as a separate subsection will make Section 2 more comprehensive.
>
> - Typos: Thank you for the careful reading. We have corrected all identified typos.
>
> **Questions**
> - Latency motivation: In LoRA, the two additional CUDA kernel calls caused by sequential matrix multiplications introduce latency overhead. This becomes more pronounced as Transformers have multiple projection matrices in each layer. (This can also be observed in Table A, comparing Base and unmerged LoRA). We would also like to clarify that $f_{in}$ and $f_{out}$ are applied only once per forward pass, not once per FFN layer. The latent embedding is enlarged once at the beginning, and this extra dimension *carries* the fine-tuned information throughout the network until it contracts at the end. Moreover, $f_{in}$ and $f_{out}$ are simple operations (concatenation, linear combinations, etc.) which are negligible in the total forward pass.
>
> - Latency profiling: We suspect the latency improvement might have appeared trivial because we did not include the latency of the base model (which serves as the lower bound for TPOT). Therefore, we included the base model latency in Table A and measured the breakdown of the components. LatTE's overhead is concentrated in attention due to the operations required in managing the multi-heads (as in "more" and "wider" heads), while LoRA does not have that distinction but incurs overhead from the FFN.
>
> - Architectural overhead: The additional projections from attention do not contribute much to the latency as they change the $d \times d$ matrix multiplied by a $d$ vector, to a $(d+r) \times (d+r)$ matrix multiplied by a $d+r$ vector. In most cases $d \gg r$, and the overhead for this operation is small. For "more heads," additional inner product, softmax, and concat operations result in latency overhead. For "wider heads," interleaving the keys and queries among heads causes increased latency. From Table A, one observes that the wider head overhead is greater, most likely because the interleaving operation is much less optimized. Please let us know if we misunderstood the projection you mentioned; we will be happy to further clarify.
>
> - Initialization strategy: This is a great point, as initialization is indeed important in these methods. We used Kaiming-uniform for $A$ and zeros for $B$ and $C$. This choice ensures that LatTE initially produces an identical forward pass to the base model, and it is also the default implementation of LoRA in the Huggingface PEFT library. Apart from initializing results of the base model, the reason we chose this initialization is for a fair comparison with LoRA. As initialization is crucial in these methods, we wanted to ensure an identical environment for comparison without over-searching the initialization space. We will clarify this in the revised manuscript.

---

### Author Response · Authors · 2025-11-27
**General Response**

We sincerely thank all reviewers for their thoughtful and constructive feedback. The reviews have helped us clarify our contribution and strengthen our experimental validation. We address the common themes that emerged across reviews below:

**Core Contribution Clarification**

Multiple reviewers noted that performance improvements over LoRA are modest (2kbB, hCrW, NXHC). We acknowledge this and clarify: our contribution is not superior accuracy but achieving LoRA-equivalent accuracy with superior multi-adapter inference efficiency. LoRA already approaches full fine-tuning performance, leaving limited room for improvement. Our focus is on deployment scenarios where multiple adapters must coexist -- personalized models, multi-domain serving, and edge devices -- where existing PEFT methods face significant efficiency challenges.

**Comprehensive Latency Analysis**

Several reviewers requested detailed latency profiling (2kbB, xVq1, NXHC). We provide comprehensive measurements in **Table A** below, which demonstrates LatTE's key advantage: constant-time inference regardless of adapter count, with efficient multi-task batch processing.

**Table A: Inference Latency Breakdown by Component**
Time per output token (in milliseconds) on Qwen2.5-7B with H100 GPU, context length 10k, averaged over 10 runs.

| Method | Split | Base (=merged LoRA) | unmerged LoRA | LatTE-m | LatTE-w |
|---|---|:---:|:---:|:---:|:---:|
| Single example | All | 0.236 | 0.294 | 0.243 | 0.264 |
|  | Attn. | 0.126 | 0.168 | 0.133 | 0.154 |
|  | FFN | 0.051 | 0.066 | 0.053 | 0.053 |
|  | Emb. | 0.059 | 0.060 | 0.057 | 0.057 |
| 8 Batch (1 task) | All | 0.229 | 0.285 | 0.238 | 0.254 |
|  | Attn. | 0.138 | 0.174 | 0.145 | 0.160 |
|  | FFN | 0.047 | 0.062 | 0.047 | 0.047 |
|  | Emb. | 0.044 | 0.049 | 0.046 | 0.047 |
| 8 Batch (4 tasks) | All | - | - | 0.247 | 0.251 |
|  | Attn. | - | - | 0.153 | 0.155 |
|  | FFN | - | - | 0.049 | 0.050 |
|  | Emb. | - | - | 0.045 | 0.046 |

Note: Merged and unmerged LoRA cannot efficiently handle heterogeneous multi-task batches without constant adapter loading/unloading.

**Enhanced Experimental Validation**

To address concerns about model scale (hCrW) and methodological rigor (xVq1, NXHC), we provide additional experiments:

- **Larger models**: Qwen2.5-14B results
- **Rank sensitivity**: r ∈ {8, 16, 32} showing consistent LatTE-LoRA parity
- **Quantization compatibility**: INT8 results showing robust performance

These additions demonstrate LatTE's effectiveness across 1-14B parameters, multiple ranks, and quantized settings.

**Theoretical Foundations**

Reviewer xVq1 requested clarification on theoretical assumptions. Our Theorems 1-2 establish equivalence between LatTE and LoRA in specific settings (linear models, attention mechanisms). While complete characterization for deep non-linear networks remains challenging -- a limitation shared by existing PEFT methods -- our comprehensive empirical validation across many configurations (2 model families, 1-14B parameters, 3 task types, 3 ranks) provides strong evidence of comparable expressive capacity.

**Paper Revisions**

Based on reviewer feedback, we will upload a revised manuscript as soon as possible. We believe these revisions substantially address all reviewers' concerns and strengthen the paper's contribution. We provide detailed responses to individual reviewer questions below and are happy to discuss any remaining concerns.

---

### Meta-Review · Area_Chair_XnkZ · 2025-12-21

**Summary:**

This paper introduces LatTE, a parameter-efficient fine-tuning method that introduces auxiliary latent embeddings to adapt pre-trained models, aiming to reduce inference latency compared to existing approaches like LoRA. While reviewers acknowledge the method's clear presentation, straightforward implementation, and broad experimental coverage across language and vision tasks, significant concerns about novelty, impact, and empirical rigor necessitate rejection.

The core criticisms are threefold. First, the novelty is limited: the central idea of widening model layers via added embeddings is recognized as a straightforward extension of prior techniques (e.g., Net2Net, LLaMA-Pro), and the related work discussion inadequately covers recent sparsity-based PEFT methods. Second, the empirical improvements are marginal: performance gains over LoRA are consistently small (often <0.5%) and unsupported by statistical significance tests or confidence intervals, failing to demonstrate a clear advantage. Third, the key claim of latency reduction is unsubstantiated: detailed profiling (e.g., breakdowns by component, cache effects) is missing, and the analysis overlooks that merged LoRA has no inference overhead, a critical baseline. Additional weaknesses include the lack of scalability experiments on very large models, insufficient theoretical grounding, and no ablation studies on hyperparameter sensitivity. While the concept is practical and well-executed, the paper does not sufficiently advance the state-of-the-art to warrant acceptance, even if the authors have provided rebuttal.

**Reviewer Concerns:**

I think some of concerns are still outstanding, and some of them were addressed already, including limited novelty, marginal empirical improvements, and unsubstantiated latency reduction.

**Reviewer Scores:**

Most of reviewers will maintain the current scores.

---

### Decision · Program_Chairs · 2026-01-26

Reject